# Identifying the Best Strategies for Improving and Developing Sustainable Rain-Fed Agriculture: An Integrated SWOT-BWM-WASPAS Approach

**Ali Firoozzare [1,*](ID), Sayed Saghaian [2,*](ID), Sasan Esfandiari Bahraseman [1] and Maryam Dehghani Dashtabi [1]**

[1] Department of Agricultural Economics, College of Agriculture, Ferdowsi University of Mashhad, Mashhad 9177948978, Iran

[2] Department of Agricultural Economic, College of Agriculture, Food and Environment, University of Kentucky, Lexington, KY 40536, USA

[*] Correspondence: firooz@um.ac.ir (A.F.); ssaghaian@uky.edu (S.S.)

**Abstract:** The practice of rain-fed agriculture plays a vital role in both the economy and food security, yet it is subject to various challenges such as climate change and institutional barriers. This study employs the Strengths, Weaknesses, Opportunities, and Threats (SWOT) analyses, Best-Worst Method (BWM), and Weighted Aggregated Sum Product Assessment (WASPAS) integrated approach to identify the most effective strategies for improving and developing sustainable rain-fed agriculture in Mashhad, Iran. The SWOT analysis identifies the essential sub-factors for improving and developing sustainable rain-fed agriculture. Then, the BWM method is utilized to assign weights to each sub-factor. Finally, the WASPAS method is used to rank the 19 strategies that can help achieve sustainable rain-fed agriculture. The findings of this study reveal that the strategy of establishing an institutional framework to promote sustainable rain-fed agriculture (WT7) has received the highest score. On the other hand, strategies related to supporting policies at the farm level (ST2, WO3, WT2, WT1) were placed in the middle and final priorities. Thus, it is recommended that in the current context of rain-fed agriculture in Mashhad, policymakers prioritize institutional policies related to rain-fed agriculture over farm-level policies. This study proposes a comprehensive and systematic approach to enhance and promote sustainable rain-fed agriculture.

**Keywords:** sustainable rain-fed agriculture; water governance; water shortage; strategy

## 1. Introduction

Agricultural production needs to be increased by 60–110% by 2050 to meet the world's increasing food demand [1]. The implementation of such a plan is considered as one of the greatest challenges facing humankind in the years to come, placing more pressure on resources such as water, land, and energy [2,3]. Meanwhile, from among the different dimensions regarding this challenge, accessing water for food production would be of the greatest concern [4,5]. According to forecasts, approximately 65 countries accommodating more than seven billion people would face severe water shortages by 2050. Moreover, about 1.2 billion people live in areas where severe lack of water and frequent occurrence of droughts have caused many problems for this type of agriculture [6,7]. As rain-fed farming is the most typical form of agriculture in these areas, water shortages cause more critical problems [8].

On the other hand, 80% of the world's agricultural lands are rain-fed, providing nearly 60% of the world's food needs, which in turn greatly helps the farmers earn a living [9]. However, while the significance of rain-fed agriculture seems to be area-restricted, it still provides the largest share of food supplies for poor people living in the developing countries [10]. For instance, 75% of the arable lands in North Africa and the Near East,

about 90% of the agricultural lands in Latin America, 95% of the farming lands in sub-Saharan Africa, and nearly 60% of the cultivated lands in South Asia are potentially capable of being rain-fed. Therefore, rain-fed agriculture has enormous potential to meet the food demands of the world's growing population [7].

However, there are various risks and threats present in rain-fed areas, including climate change, alterations in soil properties, and topographic and human changes, bringing about many restrictions and problems in rain-fed areas. In this regard, rain-fed agriculture is highly sensitive to climate change, especially air temperature and precipitation [11]. Moreover, periodic droughts may lead to socio-economic consequences for the farmers whose revenues depend on rain-fed agriculture [12]. Furthermore, the capacities of rain-fed lands for crop production are normally underestimated, with most pecuniary resources being allocated to the development of irrigated lands [13]. Considering the direct influence of weather conditions on the availability of water, rain-fed agriculture often delivers poor performance [14]. Many farmers still apply outdated unsustainable methods to rain-fed agriculture, thus decreasing the production potential of this type of farming [15].

The potential of rain-fed areas can be improved by adopting appropriate techniques and sustainable agricultural practices. Hence, to create a sustainable rain-fed system that is economically affordable, ecologically flexible, and equitable in terms of development processes, it is necessary to identify the effective factors involved in improving and developing sustainable rain-fed agriculture. Thus, this study sought to detect the best strategies for developing sustainable rain-fed agriculture based on the consideration of relevant comprehensive factors.

Several studies have attempted to identify the factors involved in rain-fed agriculture [5,7,16–21], suggesting different strategies such as determining the optimal amount of fertilizer consumption [19], product insurance [20], improving mechanization [5] and applying smart farming practices [18], etc. In this regard, the studies used various quantitative and qualitative methods, including econometrics, artificial intelligence, laboratory studies, and review of the related literature. However, the methods failed to consider all the relevant effective factors in rain-fed agriculture and their interdependence.

Therefore, this study attempted to fill the gaps in determining the strategies needed for the development and improvement of sustainable rain-fed agriculture using an integrated method based on the Strengths, Weaknesses, Opportunities, and Threats (SWOT), Best-Worst Method (BWM), and Weighted Aggregated Sum Product Assessment (WASPAS). It should be noted that this is the first study to use such a proposed approach to identify and prioritize the factors involved in improving and developing strategies concerning sustainable rain-fed agriculture.

As a useful technique for solving problems concerning strategic planning, the SWOT analysis helps decision-makers explore many areas related to the successful development of an appropriate program by identifying relevant internal and external issues [22,23]. The factors identified via SWOT analysis are then weighed using the BWM, which is more appropriate than other weighting methods as it requires fewer comparative data, providing a more stable and reliable comparison [24]. Moreover, the BWM approach can be used, together with the SWOT analysis, to evaluate and compare the relationship between different factors [25]. Finally, the WASPAS method is used to prioritize the best strategies for improving and developing sustainable rain-fed agriculture. It should be noted that similar hybrid methods have already been used to identify and prioritize strategies in different fields. For instance, Aghasafari [22], and Esfandiari [23] used SWOT-FANP and SWOT-AHP-FTOPSIS methods for the development of organic farming and water resource management, respectively.

This study contributes to the existing literature in four important ways. First, this is the first study to examine all relevant factors involved in the development and improvement of sustainable rain-fed agriculture, including political, social, economic, environmental, natural, and technical ones. Second, the strategies are selected and ranked based on the strengths, weaknesses, threats, and opportunities regarding the improvement and

development of rain-fed agriculture. Third, the interdependence of various relevant factors is taken into account. Finally, Mashhad city (located in Khorasan Razavi, Iran) was selected as the study area because of its unique conditions and characteristics compared to other sites investigated in previous studies. The remainder of this article is organized as follows. Section 2 outlines the materials and methods used in the study, which includes the study area, statistical population, and methodology. Section 3 presents the results and discussion. Finally, Section 4 is the conclusion, which provides a summary of the main findings.

## 2. Materials and Methods

### 2.1. The Study Area

Located in a dry and semi-arid region in northeastern Iran (Figure 1), Mashhad holds 83,274 hectares of agricultural lands (including 58,862 hectares irrigated and 25,412 hectares of rain-fed lands), having great capacities to be used for exploiting water and soil resources [26]. The region's required water for agricultural activities is supplied from underground and surface resources. However, while possessing 6086 deep and semi-deep wells, 656 springs, 1084 qanats, and three dams, the city suffers from drought and water shortages [27].

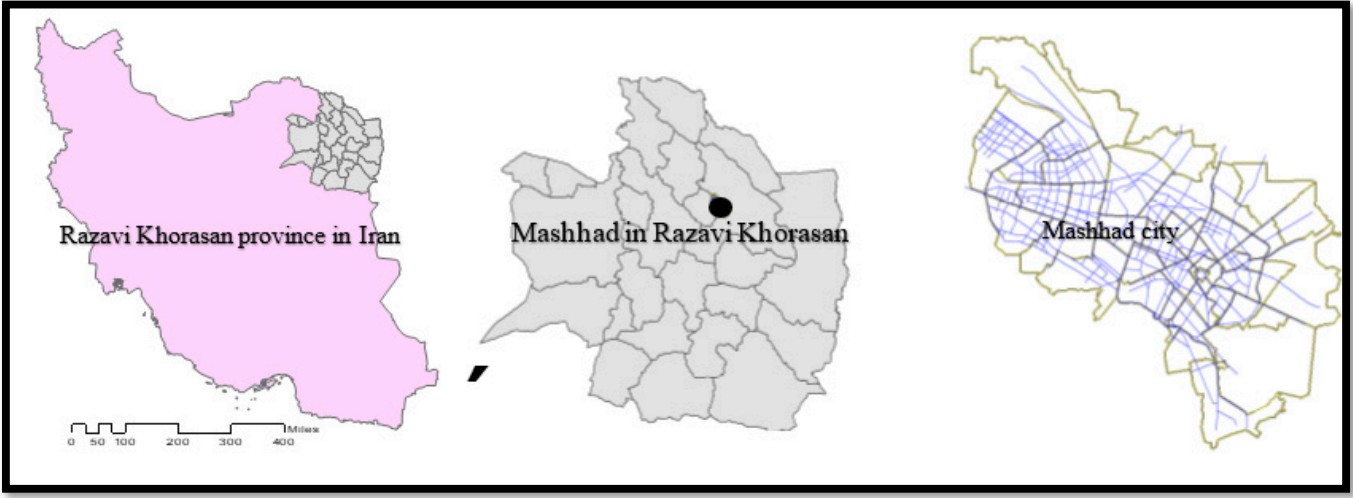

**Figure 1.** The geographical location of the study area in Khorasan Razavi province, Iran.

According to the Iranian Sixth Development Plan, the government has adopted policies to constrain the exploitation of underground water resources [28]. However, such policies may threaten food security. On the other hand, the capacity of food production could be increased by developing rain-fed agriculture without additional consumption of water resources. However, due to climate change, lack of attention to rain-fed agriculture, and poor performance of this type of farming, the areas under rain-fed agriculture are decreasing annually in Iran. Therefore, the Iranian government is seeking to lower the risks of rain-fed agriculture, making it consistent with sustainable development, and thus improving food security. In this regard, it is important for policymakers in this region to know:

The strengths, weaknesses, opportunities, and threats related to the improvement and development of rain-fed agriculture.

The strategies for improving and developing sustainable rain-fed agriculture.

### 2.2. Statistical Population

The sampling method in this study was "Sampling to Achieve Representativeness or Comparability". This method is one of the Purposive sampling methods. Purposive sampling, also known as purposeful or qualitative sampling, involves the purposeful selection of samples for acquiring knowledge or information. This type of sampling does not focus on developing fixed and immutable rules or generalizing results but rather seeks

to better understand the phenomenon in a specialized field. In this sampling, based on the mental processes, researchers determine the sample size for their study. In qualitative research, participants are selected with the aim of acquiring maximum information about the phenomenon under study [29].

This study used field studies, literature review analysis, interviews, and questionnaires as its research tools. Accordingly, nineteen interviews were made with beneficiary groups concerning the challenges that lie in improving and developing sustainable rain-fed agriculture. The interviewees included the academics who represented the scientific community, exemplary farmers, and managers and employees of governmental organizations (Table 1). The respondents were asked to express their views regarding the strategies for the improvement and development of rain-fed agriculture. Finally, five experts were requested to fill in a structured questionnaire developed by the researchers of this study.

**Table 1.** Frequency of beneficiaries' participation in semi-structured interviews.

| Participants | Number of Participants |
| --- | --- |
| Managers and employees of governmental Organizations | 7 |
| Academia | 7 |
| Exemplary farmers | 5 |
| Total | 19 |

*2.3. Methodology*

Figure 2 shows the SWOT-BWM-WASPAS integrated method used in the present study to identify the factors involved in rain-fed agriculture. The method seeks to define and prioritize alternative strategies for sustainable development of such kinds of farming.

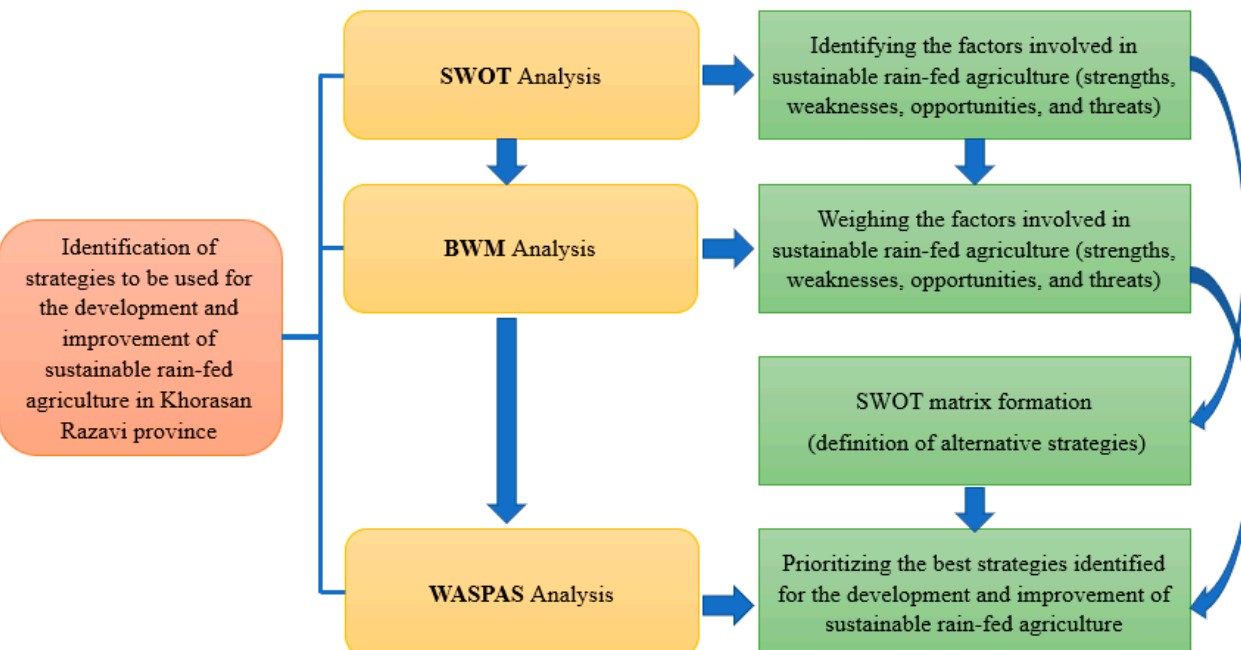

**Figure 2.** SWOT-BWM-WASPAS Framework used in this study.

Factors affecting rain-fed agriculture were identified by SWOT analysis. Then, the weight of each SWOT sub-factor was determined using the BWM approach, and alternative solutions were prioritized using the WASPAS method. In the following, the methods used are briefly presented.

### 2.3.1. SWOT Analysis

The SWOT matrix is a tool used for recognizing the threats and opportunities in the external and the weaknesses and strengths in the internal environments, helping the researchers assess the status and develop the required strategies to improve the system. In this regard, four different types of strategies are developed by combining internal and external factors within the framework of the IFE and EFE matrix. Accordingly, a suitable strategy maximizes strengths and opportunities and minimizes weaknesses and threats [30]. To this end, the strengths, weaknesses, opportunities, and threats are linked as follows in terms of four general states of WT, ST, WO, and SO, as shown in Figure 3, selecting alternative strategies accordingly [31]:

SO, Strategy (offensive): the focus is on internal strengths and external opportunities.
ST strategy (competitive): the focus is on internal strengths and external threats.
WO strategy (conservative): taking advantage of external opportunities, this strategy seeks to improve on weaknesses.
WT strategy (defensive): This strategy deals with internal weaknesses and external threats.

| Internal / External | Strengths | Weaknesses |
|---|---|---|
| Opportunities | SO Strategy (offensive) | WO strategy (conservative) |
| Threats | ST strategy (competitive) | WT strategy (defensive) |

**Figure 3.** The Structure of the SWOT Matrix.

### 2.3.2. Best-Worst Method (BWM)

As a multi-criteria decision-making method (MCDM) which is based on the matrix of pairwise comparisons, the BWM method is used to improve the decision-making process. Requiring fewer comparative data and providing more reliable answers, the method is regarded as superior to other decision-making methods [24].

STEPS of the BWM Method

Step (1) Determining the set of decision-making criteria: in this step, the decision maker is required to identify n criteria ($C_1$, $C_2$, ..., $C_n$) to be used for decision-making.

Step (2) Determining the best criterion (for instance, the most desirable or the most important criterion) and the worst one (the least desirable or the least important): in this step, the decision maker determines the best and worst criteria in general. No comparison is made at this stage.

Step (3) Determining the priority of the best criterion over other criteria, assigning values from 1 to 9. The results of the best vector in comparison with other vectors will be as follows:

$$A_B = (a_{B1}, a_{B2}, \ldots, a_{Bn}) \tag{1}$$

where $a_{Bj}$ indicates the degree of priority of the best criterion $B$ in comparison with criterion $j$. Therefore: $a_{BB} = 1$.

Step (4) Determining the degree of priority (preferability) of all criteria compared to the worst criterion assigning values from 1 to 9. The results of the superior vector in comparison with the worst one will be as follows:

$$A_w = (a_{1w}, a_{2w}, \ldots, a_{nw})^T \tag{2}$$

where $a_{jw}$ indicates the degree of priority of criterion $j$ compared to the worst criterion. Therefore: $a_{ww} = 1$.

Step (5) Finding the optimal weights: at this stage, the optimal weight of the criteria is calculated. The optimal weight of the criteria is shown as follows:

$$w_1^*, \; w_2^*, \; \ldots, \; w_n^* \tag{3}$$

The optimal weight of the criteria will be the one where the following relationships are established for each pair of $w_B/w_j$ and $w_j/w_w$:

$$\frac{w_B}{w_j} = a_{Bj} \quad , \quad \frac{w_j}{w_w} = a_{jw} \tag{4}$$

To estimate such conditions for all "$j$" items, a solution must be found to minimize the maximum absolute value of the following difference for all "$j$" items.

$$\left| \frac{w_B}{w_j} - a_{Bj} \right| \quad , \quad \left| \frac{w_j}{w_w} - a_{jw} \right| \tag{5}$$

Considering the condition that the sum of the weights should be equal to one and that the weights should not be negative, the issue will be solved as follows:

$$\begin{aligned} &\min \max_j \left\{ \left| \frac{w_B}{w_j} - a_{Bj} \right|, \left| \frac{w_j}{w_w} - a_{jw} \right| \right\} \\ &s.t. \\ &\sum_j w_j = 1 \\ &w_j \geq 0, \; for \; all \; j \end{aligned} \tag{6}$$

where the model or issue can be converted into the following problem:

$$\begin{aligned} &\min \xi \\ &s.t. \\ &\left| \frac{w_B}{w_j} - a_{Bj} \right| \leq \xi, \; for \; all \; j \\ &\left| \frac{w_j}{w_w} - a_{jw} \right| \leq \xi, \; for \; all \; j \\ &\sum_j w_j = 1 \\ &w_j \geq 0, \; for \; all \; j \end{aligned} \tag{7}$$

By solving the equation, the optimal values for the indicators' weight and the optimal value of the objective function are obtained [24].

### 2.3.3. WASPAS Technique

Regarded as an excellent multi-criteria decision-making method for selecting the best alternatives, the WASPAS was first presented by Zavadskas [32] as a combination of two models, that is WSM (weighted sum model) and WPM (Weighted product model), being more accurate compared to independent methods [32]. The steps followed in this method are as follows:

1.　Normalizing the decision matrix using Equations (8) and (9).

$$\overline{X}_{ij} = \frac{X_{ij}}{\max_i X_{ij}} \quad for \; beneficial \; criteria \tag{8}$$

$$\overline{X}_{ij} = \frac{\min_i X_{ij}}{X_{ij}} \quad for \; non-beneficial \; criteria \tag{9}$$

2.　Calculating the relative significance of the alternatives based on the WSM method using Equation (10) (this equation is the weighted matrix where the normal matrix is multiplied by the weights of the criteria).

$$Q_i^{(1)} = \sum_{j=1}^{n} \overline{X}_{ij} W_j \tag{10}$$

$$Q_i^{(2)} = \prod_{j=1}^{n} (\overline{X}_{ij})^{W_j} \tag{11}$$

3.　Calculating the relative significance of the alternatives based on the WPM method via Equation (11) (this equation also states that the normal matrix must acquire the power equal to the weights of the criteria).

4.　Calculating the common criterion, where the significance of the alternatives is calculated with equal proportion using Equations (10) and (11).

$$Q_i = 0.5 Q_i^{(1)} + 0.5 Q_i^{(2)} = 0.5 \sum_{j=1}^{n} \overline{X}_{ij} W_j + 0.5 \prod_{j=1}^{n} (\overline{X}_{ij})^{w_j} \tag{12}$$

While the alternatives can be ranked based on the $Q_i$ value, the accuracy and efficiency of the WASPAS method help calculate the relative significance of the *i*-th alternative by calculating the Landau through the following formula:

Moreover, the WASPAS method comprises a more generalized equation to determine the relative significance of the entire *i*-th alternative, thus increasing the accuracy and efficiency of the ranking of the decision-making process, the formula of which is as follows:

$$Q_i = \lambda Q_i^{(1)} + (1-\lambda) Q_i^{(2)} = \lambda \sum_{j=1}^{n} \overline{X}_{ij} W_j + (1-\lambda) \prod_{j=1}^{n} (\overline{X}_{ij})^{w_j}, \; \lambda = 0, \ldots, 1 \tag{13}$$

The following formulas are used to calculate the optimal $\lambda$ based on standard deviation.

$$\lambda = \frac{\sigma^2(Q_i)}{\sigma^2(Q_i) + \sigma^2(Q_i)} \tag{14}$$

$$\sigma^2(Q_i) = \sum_{j=1}^{n} x_{ij} w^2 \sigma^2(X_{ij}) \tag{15}$$

$$\sigma^2(Q_i) = \sum_{j=1}^{n} \left[ \frac{n(X_{ij})^{w_j} \times w_{ij}}{(X_{ij})^{w_j} (X_{ij})^{(i-w_j)}} \right] \sigma^2 \tag{16}$$

$$\sigma^2(X_{ij}) = (0.05 X_{ij})^2 \tag{17}$$

Currently, candidate alternatives are ranked based on their $Q$ values, with the alternative containing the highest $Q$ value being considered as the best alternative [33].

### 3. Discussion and Results

*3.1. The Results of the SWOT-BWM Analysis*

In the first step, internal (Table 2) and external (Table 3) factors concerning the development and improvement of sustainable rain-fed agriculture were identified by reviewing the related literature, conducting interviews, and carrying out field studies. Then, the 23 identified factors, including six strengths, four opportunities, seven weaknesses, and six threats were prioritized using the BWM. Table 2 shows the results of the prioritization of internal factors, including strengths and weaknesses concerning rain-fed agriculture. Accordingly, from among the six strengths, the S3, that is, "unnecessity of irrigation", was identified as the first priority with a weight of 0.309. According to the experts interviewed in this study, it can be argued that the feasibility of practicing rain-fed agriculture under the current status of water shortage in Mashhad could be considered as one of the most significant strengths of using rain-fed agriculture in the city. The studies conducted by Rockstrom [34], Swatuk [35], and Dunkelman [36] highlight the feasibility of rain-fed agriculture in conditions of water scarcity, as opposed to irrigated agriculture. These findings support the results of the present study. On the other hand, out of the seven prioritized weaknesses, the lack of specialized technologies for rain-fed agriculture (W7) (with a weight of 0.309) was identified as the most important weakness concerning rain-fed agriculture. As expressed by the experts interviewed in this study, available agricultural technologies are imposed on rain-fed areas without considering the special requirements of rain-fed farming. Therefore, these technologies are not suitable for rain-fed agriculture. Baig [15] highlighted the importance of innovative technologies that are appropriate for rain-fed areas in their research conducted in Pakistan.

**Table 2.** Evaluation matrix of internal factors involved in rain-fed agriculture.

| SWOT Factors | | SWOT Sub-Factors | Weight |
|---|---|---|---|
| Strengths (S) | S1 | The healthiness of rain-fed products | 0.103 |
| | S2 | Economical nature of rain-fed agriculture | 0.154 |
| | S3 | Unnecessity of irrigation | 0.309 |
| | S4 | Historical background of rain-fed agriculture in Khorasan Razavi province | 0.164 |
| | S5 | The existence of potential lands for rain-fed agriculture in Khorasan Razavi province | 0.108 |
| | S6 | Sustainable exploitation of water and soil resources | 0.163 |
| Weaknesses (W) | W1 | The farmers' lack of knowledge and skills in rain-fed agriculture | 0.093 |
| | W2 | Low performance of rain-fed agriculture | 0.078 |
| | W3 | Low marketability of rain-fed products | 0.112 |
| | W4 | Low variety of crops in rain-fed agriculture | 0.200 |
| | W5 | Sensitivity to the cultivation time | 0.072 |
| | W6 | Smallholder farmers | 0.217 |
| | W7 | Lack of rain-fed-agriculture-savvy technologies | 0.229 |

**Table 3.** Evaluation matrix of external factors regarding rain-fed agriculture.

| SWOT Factors | | SWOT Sub-Factors | Weight |
|---|---|---|---|
| Opportunities (O) | O1 | Large market of agricultural products in Iran | 0.116 |
| | O2 | The presence of an active private sector in Khorasan Razavi province | 0.115 |
| | O3 | Increasing demand for food | 0.298 |
| | O4 | The high diversity of drought-resistant native plant species | 0.472 |
| Threats (T) | T1 | Climate change and drought | 0.081 |
| | T2 | Improper time distribution of rain | 0.092 |
| | T3 | Disproportionate governmental support | 0.195 |
| | T4 | Plant pests and diseases | 0.089 |
| | T5 | Misconceptions concerning rain-fed ecosystems | 0.425 |
| | T6 | Lack of rain-fed agriculture research centers | 0.118 |

Table 3 shows the results of the prioritization of external factors concerning rain-fed agriculture, including opportunities and threats. Accordingly, out of four identified opportunities, the high diversity of drought-resistant native plant species (O4) was found to be the first priority, with its weight being 0.472. Some 6417 plant species originate in Iran, with a large number of such native species naturally existing in the country's vegetation area, including almonds, grapes, pistachios, jujubes, olives, wild olives, pomegranates, figs, and mulberries, most of which are drought resistant. The potential for the growth of native plants resistant to drought in Mashhad has been confirmed by studies conducted by Kazemi and Abbassi [37] and Kazemi [38]. On the other hand, misconception concerning the rain-fed ecosystems (T5) was identified as the most important threat, with its weight being 0.241. According to the experts interviewed in this study, dryland management has been influenced by a long history of misconceptions regarding such ecosystems, which are mainly considered as barren landscapes of little economic value due to their unfavorable climate, low productivity, and distance from markets and political centers. The current study's findings are consistent with previous research conducted by Mortimore [39] and Hoover [40] which have identified the prevalence of misconceptions as one of the most significant threats to dry ecosystems.

As shown in Figure 4, all rain-fed agriculture-related SWOT sub-factors (strengths, weaknesses, opportunities, and threats) were compared. The high diversity of drought-resistant native plant species (O4), a misconception regarding rain-fed ecosystems (T5), and the feasibility of operating rain-fed agriculture under water shortage conditions (S3) were identified as the most significant factors, with their weights being 0.472, 0.425, and 0.309, respectively.

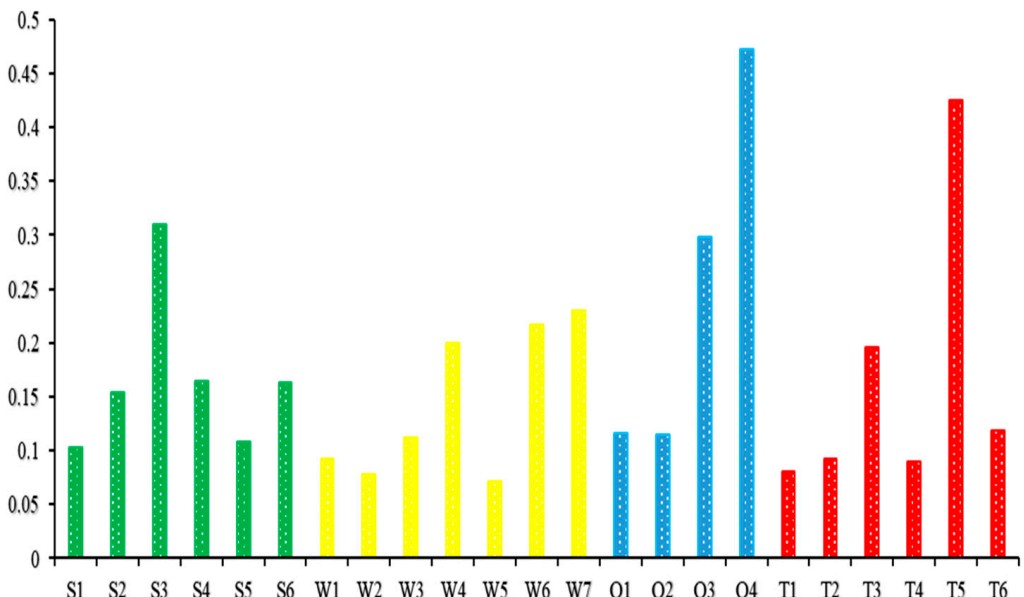

**Figure 4.** Overall ranking of SWOT's sub-factors.

### 3.2. WASPAS Results

Following the identification of the internal and external factors, the strategies required for the improvement and development of sustainable rain-fed agriculture were determined (third column of Table 4), which were then ranked using the WASPAS method (the BWM was also used to determine the weights of the criteria (SWOT sub-factors)). As shown in the second column of Table 4, this study developed nine defensive, three offensive, three competitive, and four conservative strategies. The values of λ and Qi and the ranking of the strategies concerning the development and improvement of sustainable rain-fed agriculture in Mashhad are presented in the fourth, fifth, and sixth columns of Table 4, respectively.

**Table 4.** Prioritized strategies for improvement and development of sustainable rain-fed agriculture.

| No | | Strategy | λ | Qi | Rank |
|----|-----|----------|------|------|------|
| 1 | SO1 | Using indigenous knowledge in dry farming | 0.597 | 0.136 | 19 |
| 2 | WT1 | Using seeds resistant to drought, salinity, and pests | 0.782 | 0.175 | 14 |
| 3 | SO2 | Promoting integrated agricultural systems in rain-fed areas | 0.787 | 0.196 | 11 |
| 4 | WO1 | Teaching and promoting the principles of rain-fed agriculture | 0.798 | 0.220 | 6 |
| 5 | SO3 | Using drought-resistant native plant species | 0.783 | 0.195 | 13 |
| 6 | ST1 | Developing water protection services and improving water consumption efficiency | 0.753 | 0.200 | 10 |
| 7 | ST2 | Socio-cultural intervention to raise awareness regarding rain-fed cultivation and arid lands | 0.778 | 0.211 | 9 |
| 8 | WT2 | Subsidizing inputs and guaranteed purchase of rain-fed agricultural products | 0.505 | 0.166 | 16 |
| 9 | WT3 | Improving the conditions of comprehensive insurance for rain-fed agriculture | 0.797 | 0.196 | 12 |
| 10 | SO4 | Encouraging the private sector to invest in rain-fed agriculture | 0.561 | 0.224 | 5 |
| 11 | WT4 | Protection of water and soil using conservation agriculture methods | 0.574 | 0.151 | 17 |
| 12 | WO2 | Use of new technologies suitable for sustainable rain-fed agriculture | 0.671 | 0.213 | 8 |
| 13 | ST3 | Reforming water governance | 0.650 | 0.233 | 3 |
| 14 | WO3 | Providing the farmers using rain-fed agriculture with low-interest long-term banking facilities | 0.652 | 0.196 | 15 |
| 15 | WO4 | Improving and expanding rain-fed agriculture research centers | 0.669 | 0.232 | 4 |
| 16 | WT5 | Dedicating a fair share of public resources to rain-fed agriculture | 0.773 | 0.248 | 2 |
| 17 | WT6 | Devising solutions to increase the efficiency of fertilizers and applying more effective ways to control weeds and pests | 0.702 | 0.146 | 18 |
| 18 | WT7 | Institutional framework to accelerate the growth of sustainable rain-fed agriculture | 0.725 | 0.275 | 1 |
| 19 | WT8 | Developing the use of information and communication technology (ICT) in rain-fed agriculture | 0.700 | 0.218 | 7 |

According to the results presented in Table 4, the most important strategies for the development and improvement of sustainable rain-fed agriculture in Mashhad are as follows:

WT7 Strategy: Rank 1

As multifaceted issues underlie rain-fed agriculture, this type of farming requires a comprehensive institutional framework at different levels, including national, regional, and regional scales, so that it can be directed toward sustainable solutions [41].

WT5 Strategy: Rank 2

The full realization of rain-fed agriculture's capacities requires a fair and proportionate share of public investments.

ST3 Strategy: Rank 3

Iranian policymaking and governance in water resources management have mainly been concentrated on irrigated agriculture, leading to limited investments and innovations concerning water management in dryland agriculture. Therefore, water management in rain-fed agriculture can be improved by reforming water governance. The role of water governance in the development of rain-fed areas is also supported by the FAO report from [8].

WO4 Strategy: Rank 4

The linear approach of transferring technology without understanding different ecological, social, economic, and environmental grounds of rain-fed areas emanates from the incompetency of research and development systems in assessing such areas. This approach is being increasingly aggravated due to the lack of sufficient support from rain-fed agriculture research centers. Therefore, proper grounds are suggested to be provided for the improvement and development of research centers on sustainable rain-fed agriculture [41].

SO4 Strategy: Rank 5

The demands for agricultural products are increasing with the growing increase in the world's population, providing a good opportunity for investing in the field of rain-fed agriculture that bears economic value. However, as farmers with limited capital cannot invest in rain-fed agriculture, it is suggested that agricultural entrepreneurs, the private sector, and foreign investors be invited to invest in this regard.

WO1 Strategy: Rank 6

Promoting rain-fed agriculture and instructing its principles play an important role in the overall process of rain-fed areas' sustainable development. Accordingly, governmental officials, policymakers and planners in rural areas, and non-governmental organizations need to be trained on the sustainable development of rain-fed regions. On the other hand, organizing professional courses, expanding promotional courses, and promoting

useful technologies for rain-fed agriculture among farmers can help develop this type of agriculture sustainably.

WT8 Strategy: Rank 7

While Information and Communication Technology (ICT) can lower the risks of rain-fed agriculture by ensuring the optimal use of resources, yielding more products, and reducing operational costs, it is rarely applied to such type of farming. Moreover, ICT can be used in several key areas in rain-fed agriculture, including rapid warning systems, precision agriculture, provision of information regarding markets, and climate-related issues [8,18]. The FAO report from [8] highlights the significance of focusing on ICT in the development of rain-fed areas.

WO2 Strategy: Rank 8

While the current agricultural technologies are not compatible with rain-fed areas, they are undesirably imposed on rain-fed agriculture without the consideration of proper plans to resolve the specific problems of the areas. Therefore, new rain-fed agriculture-compatible technologies and machines are required to be designed so that they can fulfill the needs of such farming [41].

ST2 Strategy: Rank 9

To counter the threats posed against the sustainability of drylands, a conscious change needs to be made in the viewpoints existing towards such lands. Highlighting the advantages of drylands via education and promotion can help ensure the land's sustainability. Therefore, standing against negative perceptions concerning dry lands is highly recommended. Moreover, the current plans and programs are required to be redefined to make them farmer friendly. Hoover [40] suggested that a conscious shift in perspectives towards dry ecosystems is a crucial strategy for enhancing and advancing agriculture in these areas. This finding aligns with the results of the current study.

ST1 Strategy: Rank 10

Collecting or harvesting more water and ensuring it penetrates to the root area can be very effective in water conservation. Furthermore, conserving water by increasing the plants' absorption capacity, reducing evaporation-induced losses, and applying drainage to the root zone may help increase the performance of rain-fed agriculture. Rainwater harvesting is an essential component for enhancing water supply in rural regions, minimizing the risk of floods, and alleviating the impacts of climate change [42,43]. Therefore, water and soil protection strategies such as watershed management, the use of terraces, and contour cultivation can play an effective role in protecting water and soil and improving the efficiency of water consumption. Furthermore, applying supplementary irrigation with techniques such as drip irrigation, etc. can be used to produce more crops [36].

SO2 Strategy: Rank 11

There is a limited variety of products in rain-fed areas. However, the flexibility and sustainability of the revenues earned from such areas can be improved by promoting Integrated Farming Systems (IFS) [44]. In other words, two or more products can be produced simultaneously (field crops, horticultural crops, livestock, aquaculture, chicken/duck, beekeeping, and mushroom cultivation).

WT3 Strategy: Rank 12

Rain-fed areas are faced with various risks and threats. In other words, climatic events such as rain and drought exert a direct influence on economic activities in rain-fed areas. Therefore, dry farming is highly sensitive to climate changes, especially to air temperature and precipitation rate. Moreover, periodic droughts may bring about socio-economic consequences for the farmers whose revenues depend on rain-fed agriculture. Therefore, risk management tools such as comprehensive plans for insuring products can be increased in rain-fed areas, considering the fact that they appear to be suitable alternatives for traditional insurance plans for agricultural products [8]. FAO [8] places significant emphasis on the importance of insuring rain-fed products. Moreover, the study conducted by Ward and Makhija [20] in India found that farmers were less inclined to cultivate rain-fed crops in the absence of insurance. SO3 Strategy: Rank 13.

Using drought-resistant native plant species can help develop sustainable rain-fed agriculture, some of which have already been domesticated and adapted to the local rain-fed conditions and are currently used for commercial fruit production in Iran. However, other types of such plants are yet to be adapted likewise. In the studies conducted by Brooks [45], Kazemi [38], and Hoover [40], the importance of native plants in enhancing agriculture in arid ecosystems has been emphasized.

WT1 Strategy: Rank 14

Considering the vulnerability of rain-fed agriculture to drought and pests, it is necessary for the farmers, especially the smallholder ones, to use improved high-yield seeds that are resistant to drought and pests [46]. Furthermore, as timely cultivation plays a very significant role in rain-fed agriculture [21], a suitable delivery mechanism is required to ensure cheaper and timely access to the seeds.

WO3 Strategy: Rank 15

Farmers in rain-fed areas are usually smallholders with weak financial ability who are faced with various risks. Therefore, providing such farmers with long-term low-interest banking facilities can help them purchase agricultural inputs on time and reduce the probable risks. Thus, farmers working in rain-fed areas need to be provided with banking facilities and better rural banking infrastructure [46]. Moreover, granting subsidies to water conservation technologies can help reduce the adverse consequences of drought and develop sustainable rain-fed agriculture.

WT2 Strategy: Rank 16

Considering the great significance of cultivation time in rain-fed agriculture, an appropriate delivery mechanism is required to ensure timely and affordable access to agricultural inputs. Moreover, the fluctuations in the products' prices together with the poor marketability of rain-fed products may incur considerable losses to farmers. Therefore, the government is required to take appropriate measures for implementing a guaranteed-purchase policy for rain-fed products and granting subsidies to production inputs [46].

WT4 Strategy: Rank 17

Conservation agriculture is considered to be an effective approach to overcoming drought crises, optimally managing water consumption, and compensating for the loss of soil's organic matter in rain-fed agriculture. In other words, conservation agriculture which is based on conserving water and soil can significantly contribute to increasing the performance of rain-fed lands by maintaining soil moisture through saving crop residues. To this end, it uses a variety of methods, including tillage, mulching, cover cropping, mixed cropping, forest farming, and the application of organic fertilizers (vermicompost, green manure) [16]. Accordingly, deep tillage increases water penetration, reduces runoff, and controls soil erosion. Baig [15] and Kassam [47] have also highlighted the importance of conservation agriculture in improving and developing rain-fed agriculture, which aligns with the findings of the present study.

WT6 Strategy: Rank 18

The low efficiency of fertilizers in rain-fed agriculture has already been reported by many studies. However, some scholars argue that producing more crops requires the application of fertilizers. Thus, they suggest that using fertilizers at the right time with the correct dosage and appropriate properties may help increase their efficiency, leading to the higher performance of rain-fed agriculture [19]. Moreover, controlling weeds and pests could play a crucial role in improving the performance of rain-fed farming. The studies conducted by Bhatti [48] in Pakistan, Chukalla [49] and Lamptey [16] in Africa also emphasized the importance of fertilization at the appropriate time and with the correct dose in order to improve yields in rain-fed agriculture. These findings support the results of the current study.

SO1 Strategy: Rank 19

Indigenous knowledge together with the experiences transferred to farmers from generation to generation can be a strong and efficient tool for sustainably managing the environment, soil, and water resources, and, therefore, developing sustainable rain-fed

agriculture, especially in the freak conditions of arid and semi-arid ecosystems [36]. In this regard, the creation of Bandsars (whose structures are the same as gabion), Houtak and Dagar systems, and areas known as Chaleh (pit) are some examples of Iranian indigenous knowledge used in rain-fed agriculture. Bandsars are used to collect runoff and sediments. The Houtak system acts as a reservoir and Dagar is a completely flat land that is cultivated using a flood irrigation technique. Chaleh is an area for farming where farmers reach fresh water by digging into the ground and using it for cultivation and gardening [50]. Atinmo [51] and Dunkelman [36] have supported the finding that traditional methods of water harvesting, which draw upon indigenous knowledge, have been effective in facilitating agricultural growth in arid regions of Africa.

Faced with a wide variety of problems, rain-fed lands are considered as invaluable resources in providing food security. However, performing rain-fed agriculture based on scientific methods and carrying out the above-mentioned strategies in order of priority can help improve and develop sustainable rain-fed agriculture.

## 4. Conclusions

Improving and developing sustainable rain-fed agriculture requires a complex decision-making process that involves weighing various positive and negative factors. Iran has been struggling to improve its rain-fed agriculture, and this study proposes an integrated approach that combines SWOT analysis, BWM technique, and WASPAS to identify the relevant factors, sub-factors, and prioritize strategies for sustainable rain-fed agriculture development. In this study, the SWOT factors affecting rain-fed agriculture in Mashhad were analyzed, and their weights were determined using the Best-Worst Method (BWM) technique. The study proposes 19 strategies to enhance the sustainable development of rain-fed agriculture in the region based on the proposed approach.

According to the findings of this study, the WT7 strategy, which involves creating an institutional framework to promote the growth of sustainable rain-fed agriculture, is the highest priority for improving and developing rain-fed agriculture in Mashhad. Following that, the allocation of a fair share of public resources to rain-fed agriculture (WT5) and the reform of water governance (ST3) were ranked as the second and third most important strategies, respectively. This means that to enhance and develop sustainable rain-fed agriculture in Mashhad, policymakers should initially focus on macro-level strategies related to rain-fed agriculture. Once these macro-level strategies are successfully implemented, policymakers can then shift their attention to other dimensions of rain-fed agriculture, such as the research sector, infrastructure related to rain-fed agriculture, and support policies at the farm level. The prioritized implementation of the formulated strategies can effectively assist policy makers in improving and developing sustainable rain-fed agriculture in Mashhad. In this way, the proposed decision-making approach can manage decisions' uncertainty, imprecision, and complexity as they emerge from different and conflicting criteria. Therefore, the proposed integrated approach can be used as a basis for the detailed development of comprehensive plans and policies for improving and developing sustainable rain-fed agriculture.

One of the advantages of the proposed integrated approach in this research is the possibility of combining quantitative and qualitative approaches to objectively examine the factors involved in the improvement and development of sustainable rain-fed agriculture. To further enhance this approach, future studies could incorporate new MCDM methods, such as the Ordinal Priority Approach (OPA) [52], Gray OPA [53], Fuzzy OPA [53], Robust OPA [54], and DEA-OPA [55], to conduct similar studies in different regions. By comparing the findings of these studies with each other, researchers can gain deeper insights into the effectiveness of the proposed approach and identify areas for improvement.

**Author Contributions:** Conceptualization, A.F. and S.E.B.; methodology, A.F.; formal analysis, S.S. and M.D.D.; data curation, S.E.B. and M.D.D.; writing—original draft preparation, S.S.; writing—review and editing, S.S. All authors have read and agreed to the published version of the manuscript.

**Funding:** This research received no external funding.

**Institutional Review Board Statement:** Not applicable.

**Data Availability Statement:** Not applicable.

**Conflicts of Interest:** The authors declare no conflict of interest.

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
