# Peer review of "Identifying the Best Strategies for Improving and Developing Sustainable Rain-Fed Agriculture: An Integrated SWOT-BWM-WASPAS Approach"

_agriculture, doi:10.3390/agriculture13061215_

Round 1

Reviewer 1 Report

Rain-fed agriculture makes a significant contribution to the economy and food security, but it is affected by many factors such as climate change and institutional factors. This article combines various methods to study the rain-fed agriculture in Mashhad, Iran, and puts forward some countermeasures to promote the sustainable development of rain-fed agriculture, which has certain practical and guiding value. However, some problems in the article still need to be further modified to meet the requirements of publication.

Abstract

The abstract needs to be further simplified. For example, the background can be clarified in one sentence, which does not need to be too long. Your conclusion needs to be more detailed.

Introduction

Some of the content needs to be further combed and simplified. For example, lines 77-88 and 103-110, you describe why you are doing this research and how this research contributes to other research. These two parts can be integrated because some of the descriptions are actually repeated.

The contents of Table 1 can be converted into words to further condense. Or put it into the results and discussion. Compare and discuss the contents of this part with your research results as the situation of rain-fed agriculture in other parts of the world.

Materials and methods

How do interviewees choose and what are the criteria for selection? Is the distribution of interviewees within Mashhad taken into account? In addition, is 19 interviews a little small?

The description of the methodology is suggested to be reasonably concise.

Results and discussion

The part needs to be further deepened, because only 8 references are cited in the current results and discussion, accounting for only 25% of the total references. It is suggested to add references for discussion, such as comparison and discussion with rain-fed agriculture in other parts of the world.

Reference

Some of the references were not properly formatted, some lacked volume and page numbers, some journals were not abbreviated as required, and some lacked periods between the journal names and the years. Please check it carefully.

Many of the charts in the article need to be further improved.

(1)Figure 1: It is suggested to add a picture frame. In addition, scale and legend are missing in Figure 1.

(2)Figure 2: The size of the arrows in the figure is inconsistent. If the thickness of the arrows represents a change, it needs to be explained. The block diagram is not aligned. It is recommended to adjust and optimize it to make it look Beautiful.

(3)Formula (6) and (7) are not centered like other formulas.

(4)Figure 4: Header and axis fonts do not match. Numbers and English are best written in Times New Roman. It is best to use professional drawing software to draw such as origin, which is more beautiful and professional. In addition, compared with others, there is blue boxes on the bar chart of T6. it means something else? If so, it needs to be explained.

(5)In this paper, many tables span pages, so it is suggested to adjust them so that they can be displayed on the same page as much as possible.

(6)The first letters in the table are not uniform. Some of the first letters in the same table are capitalized, while some of the letters are lowercase. For example, the first letter of the “author” and “area of study” in Table 1 should be capitalized. Similar problems exist in Table 2.

(7)The fonts in the table are not uniform. For example, Table 6: The rank column "19" is bolded, while the rest of the table text is not bolded.

Author Response

Response to Reviewer 1 Comments

Comments and Suggestions for Authors

Rain-fed agriculture makes a significant contribution to the economy and food security, but it is affected by many factors such as climate change and institutional factors. This article combines various methods to study the rain-fed agriculture in Mashhad, Iran, and puts forward some countermeasures to promote the sustainable development of rain-fed agriculture, which has certain practical and guiding value. However, some problems in the article still need to be further modified to meet the requirements of publication.

Abstract

The abstract needs to be further simplified. For example, the background can be clarified in one sentence, which does not need to be too long. Your conclusion needs to be more detailed.

Thank you for your time and consideration. The abstract underwent revisions and modifications following the reviewer's critique.

Introduction

Some of the content needs to be further combed and simplified. For example, lines 77-88 and 103-110, you describe why you are doing this research and how this research contributes to other research. These two parts can be integrated because some of the descriptions are actually repeated.

Thank you for your constructive comment. we have made revisions to the text based on your suggestions, including removing or merging the duplicated sections.

The contents of Table 1 can be converted into words to further condense. Or put it into the results and discussion. Compare and discuss the contents of this part with your research results as the situation of rain-fed agriculture in other parts of the world.

As per the suggestions of the respected reviewer, we have summarized the contents of Table 1 in words and compared them with the findings of our study.

Materials and methods

How do interviewees choose and what are the criteria for selection? Is the distribution of interviewees within Mashhad taken into account? In addition, is 19 interviews a little small?

Thank you in advance for your review. In the following text, we have provided an explanation of the reasons for selecting the interviewees, as well as the selection criteria. Furthermore, we have added a summary paragraph to the article to provide a clear and concise overview of our key findings and their implications.

Sampling in qualitative and quantitative research differs significantly in their aims, with qualitative research aiming to gain a deep understanding of the phenomenon being investigated rather than generalizing results. In qualitative research, participants are selected with the aim of acquiring maximum information about the phenomenon under study, while in quantitative research, random selection is emphasized to ensure an equal chance for all members of the research community. Purposive sampling, also known as purposeful or qualitative sampling, involves the purposeful selection of samples for acquiring knowledge or information. This type of sampling does not focus on developing fixed and immutable rules or generalizing results, but rather seeks to better understand the phenomenon in a specialized field. Purposive sampling includes three main types: "Sampling to Achieve Representativeness or Comparability", "Sampling Special or Unique Cases", and "Sequential Sampling". One of the primary challenges is determining when to stop selecting participants and how to document the decision-making process. Saturation in qualitative research is considered the gold standard method of sampling. While some researchers believe that the point of saturation is subjective, it is generally agreed that data saturation can be reached when conducting further interviews or observations does not yield new insights or modify the existing theory or perspective.

Based on the mental processes, researchers determine the sample size for their study. In their study titled "Determination of the Best Strategies for Development of Organic Farming: A SWOT –Fuzzy Analytic Network Process Approach", Aghasafari et al. (2020) conducted interviews with 20 experts. Takeleb et al. (2020) conducted 25 interviews, Balezentis et al.  (2021) conducted 21 interviews, Esfandiari et al. (2022) conducted 20 interviews, while Klager et al. (2019) conducted only 4 interviews.

Based on the aforementioned reasons, this study selected 19 interviewees. Qualitative research aims to gather comprehensive information about the phenomenon being studied, and therefore, the participants were chosen accordingly. The purpose of the study was to identify strategies for enhancing rainfed agriculture in Mashhad. As a result, the interviewees were selected from various interest groups related to rainfed agriculture, such as academics, model farmers, and government officials involved in dry farming. The interviewees were chosen to have expertise in dry farming and familiarity with the geographical, environmental, social, and economic conditions related to rainfed agriculture in the of Mashhad city.

Referencias

Hadi, Ranjbar, Haghdoost Ali Akbar, Salsali Mahvash, Khoshdel Alireza, Soleimani Mohammadali, and Bahrami Nasim. "Sampling in qualitative research: a guide for beginning." (2012): 238-250.

Aghasafari, H., Karbasi, A., Mohammadi, H., & Calisti, R. (2020). Determination of the best strategies for development of organic farming: A SWOT–Fuzzy Analytic Network Process approach. Journal of Cleaner Production277, 124039.

Takeleb, A., Sujono, J., & Jayadi, R. (2020). Water resource management strategy for urban water purposes in Dili Municipality, Timor-Leste. Australasian Journal of Water Resources, 24(2), 199-208.

Esfandiari, S., Dourandish, A., Firoozzare, A., & Taghvaeian, S. (2022). Strategic planning for exchanging treated urban wastewater for agricultural water with the approach of supplying sustainable urban water: a case study of Mashhad, Iran. Water Supply22(12), 8483-8499.

Kolagar, M. (2019). Adherence to urban agriculture in order to reach sustainable cities; a BWM–WASPAS approach. Smart Cities2(1), 31-45.

Balezentis, T., Siksnelyte-Butkiene, I., & Streimikiene, D. (2021). Stakeholder involvement for sustainable energy development based on uncertain group decision making: prioritizing the renewable energy heating technologies and the BWM-WASPAS-IN approach. Sustainable Cities and Society73, 103114.

Takeleb et al. (2020) conducted 25 interviews, Balezentis et al.  (2021) conducted 21 interviews, Esfandiari et al. (2022) conducted 20 interviews, while Klager et al. (2019) conducted only 4 interviews.

The description of the methodology is suggested to be reasonably concise.

Thank you in advance for your review and comment. As much as possible, we have removed unimportant parts of the methodology.

Results and discussion

The part needs to be further deepened, because only 8 references are cited in the current results and discussion, accounting for only 25% of the total references. It is suggested to add references for discussion, such as comparison and discussion with rain-fed agriculture in other parts of the world.

Thank you for your constructive comment. Based on the esteemed reviewer's feedback, we have added sources to the discussion section and provided comparisons.

Reference

Some of the references were not properly formatted, some lacked volume and page numbers, some journals were not abbreviated as required, and some lacked periods between the journal names and the years. Please check it carefully.

Thank you in advance for your comment. Based on the guidelines of the Journal of Agriculture, we have added references from reliable sources such as Google Scholar and Mendeley Reference Manager. However, it should be noted that the limitations identified in the study may be partly attributed to the type of article or website used as sources.

Many of the charts in the article need to be further improved.

(1) Figure 1: It is suggested to add a picture frame. In addition, scale and legend are missing in Figure 1. 

Thank you in advance for your comment and review. Based on the suggestion of the honorable reviewer, the Figure 1 in the manuscript has been revised.

(2) Figure 2: The size of the arrows in the figure is inconsistent. If the thickness of the arrows represents a change, it needs to be explained. The block diagram is not aligned. It is recommended to adjust and optimize it to make it look Beautiful.

Thank you in advance for your comment and review. Based on the suggestion of the honorable reviewer, the Figure 2 in the manuscript has been revised.

(3) Formula (6) and (7) are not centered like other formulas.

Thank you for your time and consideration. Was corrected.

(4) Figure 4: Header and axis fonts do not match. Numbers and English are best written in Times New Roman. It is best to use professional drawing software to draw such as origin, which is more beautiful and professional. In addition, compared with others, there is blue boxes on the bar chart of T6. it means something else? If so, it needs to be explained.

Thank you for your time and consideration. Was corrected.

(5) In this paper, many tables span pages, so it is suggested to adjust them so that they can be displayed on the same page as much as possible.

Thank you for your constructive comment. As the track changes feature is currently active, it may not be possible to provide a revised version of the text at this time.

(6) The first letters in the table are not uniform. Some of the first letters in the same table are capitalized, while some of the letters are lowercase. For example, the first letter of the “author” and “area of study” in Table 1 should be capitalized. Similar problems exist in Table 2.

Thank you for your time and consideration. Was corrected.

(7) The fonts in the table are not uniform. For example, Table 6: The rank column "19" is bolded, while the rest of the table text is not bolded.

Thank you for your time and consideration. Was corrected.

Reviewer 2 Report

This study tries to identify the best strategies for improving and developing sustainable rain-fed agriculture in Mashhad, Iran using SWOT-BWM-WASPAS methods. The topic is valuable, yet my comments are as follows:

-        The structure of the abstract is not suitable. The abstract should be based on the following format: 1- What is the purpose? 2- Research question 3- methodology 4- validation 5- results 6- why are your results significant?

-        The structure of the paper should be explained at the end of the introduction section.

-        Figure 2 is not readable. The authors should improve its quality.

-        Eq. (14) should be fixed.

-        The methods of the research are almost old. The authors should mention recent MCDM methods and suggest them for future studies. Some recent methods, such as the ordinal priority approach (opa), grey opa, fuzzy opa, robust opa, dea-opa can be added as the recent development of the MCDM methods.

-        The conclusion section requires a major revision.

Author Response

Response to Reviewer 2 Comments

Comments and Suggestions for Authors

This study tries to identify the best strategies for improving and developing sustainable rain-fed agriculture in Mashhad, Iran using SWOT-BWM-WASPAS methods. The topic is valuable, yet my comments are as follows:

-        The structure of the abstract is not suitable. The abstract should be based on the following format: 1- What is the purpose? 2- Research question 3- methodology 4- validation 5- results 6- why are your results significant?

Thank you for your time and consideration. The abstract underwent revisions and modifications following the reviewer's critique.

-        The structure of the paper should be explained at the end of the introduction section.

Thank you for your constructive comment. Based on the feedback of the respected reviewer, this part has been added to the article.

-        Figure 2 is not readable. The authors should improve its quality.

Thank you in advance for your comment and review. Based on the suggestion of the honorable reviewer, the Figure 2 in the manuscript has been revised.

-        Eq. (14) should be fixed.

Thank you for your time and consideration. Was corrected.

-        The methods of the research are almost old. The authors should mention recent MCDM methods and suggest them for future studies. Some recent methods, such as the ordinal priority approach (opa), grey opa, fuzzy opa, robust opa, dea-opa can be added as the recent development of the MCDM methods.

Thank you for your constructive comment. we have made revisions to the text based on your suggestions.

-        The conclusion section requires a major revision.

Thank you for your time and consideration. The conclusion underwent revisions and modifications following the reviewer's critique.

Reviewer 3 Report

1. Introduction

Page 4, lines   103-106, the authors mentioned that “This study contributes to the existing literature in four important ways. First, this is the first study to examine all relevant factors involved in the development and improvement of sustainable rain-fed agriculture, including political, social, economic, environmental, natural, and technical ones.” There are some missing factors such as rainwater harvesting which is a very important strategy that should be considered for improving the rain-fed agriculture. So it is preferred to rewrite the statement to be “This study contributes to the existing literature in four important ways. First, this is the first study to examine all relevant factors involved in the development and improvement of sustainable rain-fed agriculture, including political, social, economic, environmental, natural, and technical ones.”  

In addition, some literature should be mentioned related to rainwater harvesting. Following are good examples that could enhance the quality of this article:

Al-Batsh, N., Al-Khatib, I.A., Ghannam, S., Anayah, F., Jodeh, S., Hanbaly, G., Khalaf, B., van der Valk, M. (2019). Assessment of rainwater harvesting systems in poor rural communities: A case study from Yatta area, Palestine. Water 2019, 11, 585; doi: 10.3390/w11030585. Publisher: MDPI AG, Basel, Switzerland.

Celik, I., Tamimi, L.M.A., Al-Khatib, I.A., Apul, D.S. (2017). Management of rainwater harvesting and its impact on the health of people in the Middle East: case study from Yatta town, Palestine. Environmental Monitoring and Assessment, 189(6): 271. DOI 10.1007/s10661-017-5970-y.

2.2. Statistical Population

Table 2, in total 19 beneficiaries participated in the semi-structured interviews.

Do you think that this is a statistically representative sample that you can rely on? How? More details with references should be presented.

Author Response

Response to Reviewer 3 Comments

Comments and Suggestions for Authors

  1. Introduction

Page 4, lines   103-106, the authors mentioned that “This study contributes to the existing literature in four important ways. First, this is the first study to examine all relevant factors involved in the development and improvement of sustainable rain-fed agriculture, including political, social, economic, environmental, natural, and technical ones.” There are some missing factors such as rainwater harvesting which is a very important strategy that should be considered for improving the rain-fed agriculture. So it is preferred to rewrite the statement to be “This study contributes to the existing literature in four important ways. First, this is the first study to examine all relevant factors involved in the development and improvement of sustainable rain-fed agriculture, including political, social, economic, environmental, natural, and technical ones.”  

In addition, some literature should be mentioned related to rainwater harvesting. Following are good examples that could enhance the quality of this article:

Al-Batsh, N., Al-Khatib, I.A., Ghannam, S., Anayah, F., Jodeh, S., Hanbaly, G., Khalaf, B., van der Valk, M. (2019). Assessment of rainwater harvesting systems in poor rural communities: A case study from Yatta area, Palestine. Water 2019, 11, 585; doi: 10.3390/w11030585. Publisher: MDPI AG, Basel, Switzerland.

Celik, I., Tamimi, L.M.A., Al-Khatib, I.A., Apul, D.S. (2017). Management of rainwater harvesting and its impact on the health of people in the Middle East: case study from Yatta town, Palestine. Environmental Monitoring and Assessment, 189(6): 271. DOI 10.1007/s10661-017-5970-y.

We would like to express our gratitude to the respected reviewer for their excellent suggestion and for introducing us to valuable articles. As a result of their feedback, we have revisited our strategy ST1: Rank 10, and have considered the strategy of "Rainwater Harvesting" in greater detail. We found the articles suggested by the reviewer to be highly informative and have used the knowledge gained to develop an improved strategy. Once again, we thank the reviewer for their valuable input.

2.2. Statistical Population

Table 2, in total 19 beneficiaries participated in the semi-structured interviews.

Do you think that this is a statistically representative sample that you can rely on? How? More details with references should be presented.

Thank you in advance for your review. In the following text, we have provided an explanation of the reasons for selecting the interviewees, as well as the selection criteria. Furthermore, we have added a summary paragraph to the article to provide a clear and concise overview of our key findings and their implications.

Sampling in qualitative and quantitative research differs significantly in their aims, with qualitative research aiming to gain a deep understanding of the phenomenon being investigated rather than generalizing results. In qualitative research, participants are selected with the aim of acquiring maximum information about the phenomenon under study, while in quantitative research, random selection is emphasized to ensure an equal chance for all members of the research community. Purposive sampling, also known as purposeful or qualitative sampling, involves the purposeful selection of samples for acquiring knowledge or information. This type of sampling does not focus on developing fixed and immutable rules or generalizing results, but rather seeks to better understand the phenomenon in a specialized field. Purposive sampling includes three main types: "Sampling to Achieve Representativeness or Comparability", "Sampling Special or Unique Cases", and "Sequential Sampling". One of the primary challenges is determining when to stop selecting participants and how to document the decision-making process. Saturation in qualitative research is considered the gold standard method of sampling. While some researchers believe that the point of saturation is subjective, it is generally agreed that data saturation can be reached when conducting further interviews or observations does not yield new insights or modify the existing theory or perspective.

Based on the mental processes, researchers determine the sample size for their study. In their study titled "Determination of the Best Strategies for Development of Organic Farming: A SWOT –Fuzzy Analytic Network Process Approach", Aghasafari et al. (2020) conducted interviews with 20 experts. Takeleb et al. (2020) conducted 25 interviews, Balezentis et al.  (2021) conducted 21 interviews, Esfandiari et al. (2022) conducted 20 interviews, while Klager et al. (2019) conducted only 4 interviews.

Based on the aforementioned reasons, this study selected 19 interviewees. Qualitative research aims to gather comprehensive information about the phenomenon being studied, and therefore, the participants were chosen accordingly. The purpose of the study was to identify strategies for enhancing rainfed agriculture in Mashhad. As a result, the interviewees were selected from various interest groups related to rainfed agriculture, such as academics, model farmers, and government officials involved in dry farming. The interviewees were chosen to have expertise in dry farming and familiarity with the geographical, environmental, social, and economic conditions related to rainfed agriculture in the of Mashhad city.

Referencias

Hadi, Ranjbar, Haghdoost Ali Akbar, Salsali Mahvash, Khoshdel Alireza, Soleimani Mohammadali, and Bahrami Nasim. "Sampling in qualitative research: a guide for beginning." (2012): 238-250.

Aghasafari, H., Karbasi, A., Mohammadi, H., & Calisti, R. (2020). Determination of the best strategies for development of organic farming: A SWOT–Fuzzy Analytic Network Process approach. Journal of Cleaner Production277, 124039.

Takeleb, A., Sujono, J., & Jayadi, R. (2020). Water resource management strategy for urban water purposes in Dili Municipality, Timor-Leste. Australasian Journal of Water Resources, 24(2), 199-208.

Esfandiari, S., Dourandish, A., Firoozzare, A., & Taghvaeian, S. (2022). Strategic planning for exchanging treated urban wastewater for agricultural water with the approach of supplying sustainable urban water: a case study of Mashhad, Iran. Water Supply22(12), 8483-8499.

Kolagar, M. (2019). Adherence to urban agriculture in order to reach sustainable cities; a BWM–WASPAS approach. Smart Cities2(1), 31-45.

Balezentis, T., Siksnelyte-Butkiene, I., & Streimikiene, D. (2021). Stakeholder involvement for sustainable energy development based on uncertain group decision making: prioritizing the renewable energy heating technologies and the BWM-WASPAS-IN approach. Sustainable Cities and Society73, 103114.

Takeleb et al. (2020) conducted 25 interviews, Balezentis et al.  (2021) conducted 21 interviews, Esfandiari et al. (2022) conducted 20 interviews, while Klager et al. (2019) conducted only 4 interviews.

Round 2

Reviewer 2 Report

The authors applied my comments very well. However, there are some minor issues that should be fixed before acceptance:

  - "Regarded as an excellent multi-criteria decision-making method for selecting the best alternatives, the WASPAS was first presented by Zavadskas et al.)2016) as a combination". It seems that the style of the citation is wrong. 

- There are several typos that should be solved. 

- In conclusion, the authors mentioned the OPA and its extensions but they did not mention references. Please add related references for future readers.

Author Response

Response to Reviewer 2 Comments

"Regarded as an excellent multi-criteria decision-making method for selecting the best alternatives, the WASPAS was first presented by Zavadskas et al.)2016) as a combination". It seems that the style of the citation is wrong. 

Thank you for your time and consideration. Was corrected.

There are several typos that should be solved. 

Thank you for your time and consideration. Was corrected.

In conclusion, the authors mentioned the OPA and its extensions but they did not mention references. Please add related references for future readers.

Thank you for your time and consideration. Was corrected.

Reviewer 3 Report

The article can be accepted for publication.

Author Response

Response to Reviewer 3 Comments

The article can be accepted for publication.

Reviewer 3 has not commented to correct the revised manuscript.
